# Numerical Study on Flow and Noise Characteristics of an NACA0018 Airfoil with a Porous Trailing Edge

**Weijun Zhu** [1], **Jiaying Liu** [1,2], **Zhenye Sun** [1], **Jiufa Cao** [1,3], **Guangxing Guo** [1] **and Wenzhong Shen** [1,*]

1    College of Electrical, Energy and Power Engineering, Yangzhou University, Yangzhou 225009, China
2    Zhejiang Windey Co., Ltd., Hangzhou 310012, China
3    Department of Wind Energy, Technical University of Denmark, 2800 Kgs. Lyngby, Denmark
*    Correspondence: wzsh@yzu.edu.cn

**Abstract:** An airfoil with a porous trailing edge has a low noise emission; thus, using a porous medium is a good technique for further reduction of wind turbine noise. In this paper, to reduce airfoil trailing edge noise while minimizing the negative influence of a porous medium on aerodynamic performance, a new filling method is proposed such that a porous medium is only used in the suction side half of the trailing edge, which is more sensitive to the noise generation. The large eddy simulation (LES) technique for flow and the Ffowcs Williams and Hawkings (FW-H) method for acoustics are used. At a Reynolds number of $2.63 \times 10^5$ and various angles of attack, an NACA0018 airfoil profile with a porous trailing edge covering 20% of the chord is studied under two porous configurations, namely a fully porous and a suction-side porous trailing edge type. The results show that the flow direction, velocity magnitude, and their distributions along the boundary layer of the two porous airfoils are significantly modified due to the presence of the porous medium. The fluctuation of the pressure coefficient and the increase in the boundary layer thickness are significant at low angles of attack. As compared to the solid airfoil counterpart, the noise radiation from the newly proposed suction-side porous airfoil achieves a noise reduction of 4.3 dB at an angle of attack $\alpha = 0°$, and a noise reduction of 4.07 dB at an angle of attack $\alpha = 2°$.

**Keywords:** porous medium; NACA0018; large eddy simulation; Ffocws Williams and Hawkings (FW-H) method

## 1. Introduction

Broadband noise generated by wind turbines has negative impacts on the residents living nearby, which can limit the development and popularization of wind power generation. The total noise spectrum from a wind turbine is made up of all the aerodynamic noise mechanisms, such as the turbulent inflow noise, turbulent boundary layer trailing edge noise, flow separation noise, laminar boundary layer vortex shedding noise, trailing edge bluntness noise, and blade tip noise. Among these noise sources, the turbulent boundary layer trailing edge noise is known as the dominant aerodynamic noise source from a wind turbine and needs to be reduced. In order to reduce the turbulent boundary layer trailing edge noise, different techniques have been proposed. Studies on airfoil design and optimization have been conducted to directly decrease the noise emissions by adjusting the aerodynamic shape [1–3], such that the suction side of the boundary layer flow perturbation near the trailing edge is suppressed. The boundary layer flow injection/suction treatments at the airfoil trailing edge [4,5] have been investigated, which, combined with a control strategy, showed that a considerable decrease in noise was achieved due to the suppression and alleviation of vortices. Several methods inspired by bionics have been developed, such as serrations applied at the airfoil leading edge and trailing edge [6–8] and finlets for rotor noise reduction [9,10], and both add-ons showed good performance in noise reduction by either limiting the magnitude or spanwise correlation length-scale of the turbulence.

Among the above-mentioned contributions to airfoil noise reductions, porous material has also shown great potential for noise source reductions and noise absorption ability. To enable numerical simulations, firstly, the mathematical description and its representation of porous material in the governing equations needed to be established. As Darcy's Law reads, the porous effect can be described as the linear correlation of pressure drop with volume-averaged fluid velocity in a porous domain [11,12]. However, Darcy's Law is valid at a very small fluid velocity. Thus, further approaches have been carried out, such as the Darcy–Forchheimer Law [13,14]. By applying the source terms that represent the resistance produced by a porous material to the Navier–Stokes (NS) equations, the governing equations in a porous domain were derived [15,16], forming the basis of numerical studies. A brief instruction of the porous model implemented in the Ansys Fluent is given in the Appendix A.

On the basis of the modified NS equations, simulations and analyses on the influence of porous media on the flow field and noise generation are in progress. Naito [17] and Bhattacharyya et al. [18] have conducted simulations on a circular cylinder with applications of porous materials. Their works have illustrated that the mechanism of noise reduction lies in the damped oscillations, and the fluctuations of pressure were suppressed. Tamaro et al. [19] analyzed the far-field noise radiations of porous airfoils subjected to the turbulence shed by an upstream cylindrical rod, and the results showed that noise was reduced at low frequencies, while additional noise may regenerate at high frequencies. To deal with turbulence-impingement noise, Teruna et al. [20] identified the differences between a porous leading edge and a leading edge serration. They found that a serrated-porous leading edge exhibited an optimal noise reduction performance and a lower aerodynamic penalty. Zamani [21] employed porous media on a vertical-axis wind turbine (VAWT), in which the porous blades experienced a higher stall angle of attack, and enhancements in the power and torque coefficients in the range before and after the optimum tip speed ratio were achieved. Bernicke [22] verified the effectiveness and applicability of porous media with a single vortex passing through a porous trailing edge, which showed the desired effect on noise reduction, underlining the validity of the porous model and its application for airfoil noise reduction. Carpio et al. [23] investigated a porous NACA0018 airfoil and proposed that the reduced velocity fluctuations can be another mechanism related to low-frequency noise suppression. An encouraging finding is that a porous trailing edge with higher permeability provides up to an 11 dB noise attenuation with respect to the solid case.

In the process of different porous materials applied for noise reduction, problems have emerged as well. When a single and homogeneous porous medium is applied, a tonal noise can be generated at the boundary interface of the solid and porous parts since they possess different physical properties. Thus, optimized configurations of porous material have been studied. Zhou [24] put forward a discrete adjoint framework on trailing edge noise minimization, which was performed by determining the optimal distribution of variables governing the porosity and permeability. In order to investigate the noise suppression mechanisms, Schulze et al. [25] carried out a study by optimizing the spatial distribution of permeability based on iterative adjoint methods. Seeing that it is hard to realize the total noise reductions with only one kind of homogeneous material, a finite-element-based numerical method was proposed by Yoon [26] to achieve better total noise absorption. Another drawback of the porous treatment is the sacrificed aerodynamic performance. Aldheeb et al. [27] performed wind tunnel tests and numerical simulations and found that the slope of lift decreased while the porosity was increased, and the drag was decreased at a relatively low porosity. The negative impact on the lift was also shown in both the numerical predictions and experimental works of porous airfoils with large porosity by Hajian et al. [28].

The purpose of this study was to investigate a new filling configuration such that porous material is only installed on the suction side of the airfoil trailing edge, which is more sensitive to flow changes, with the expectation of achieving a better balance between aerodynamic force loss and noise reduction. The paper is presented as follows. In Section 2,

the flow and acoustic governing equations are presented, which include the treatment of porous media and the numerical configuration. In Section 3, the acoustical performances of solid and porous airfoils are compared, and a validation of the computational method is presented for the solid trailing edge case. Finally, the main findings are summarized in Section 4.

## 2. Computational Methodology

### 2.1. Flow Governing Equations: Modified Navier–Stokes Equations

The numerical simulation of the flow through a porous medium was achieved via a volume-averaging method. Penalty terms were added to the original NS equations, which represent the resistance exerted by the porous medium on the flow through a porous block.

In consideration of porous effects, the continuity equation for a macroscopic flow in the porous block is given by

$$\frac{\partial(\rho\varphi)}{\partial t} + \nabla \cdot \left(\rho\vec{V}\right) = 0, \ \vec{V} = \varphi\vec{v} \tag{1}$$

where $\rho$ is the density, $\vec{V}$ is the Darcy velocity, $\vec{v}$ is the intrinsic averaged velocity, and $\varphi$ is the porosity of the material.

The modification of the momentum equation for the macroscopic flow in porous media is generally based on the empirical equations obtained from experimental data. As Darcy's Law reads, a friction term can be used to model the effect of porous material, which reveals a linear relationship between the flow velocity and the pressure gradient:

$$\vec{V} = -\frac{K}{\mu}\frac{\partial p}{\partial x} \tag{2}$$

where $\mu$ is the dynamic viscosity, and $K$ is the permeability.

However, Darcy's Law is only applicable for generic steady flows at low Reynolds numbers and gives a reliable modeling of laminar flows in porous material. Equation (2) can no longer provide a suitable numerical treatment when it develops into turbulent flows with increased velocity; in such a case, an extra non-linear term is added in order to better describe the turbulent flow characteristics. Considering that the inertial force increases continuously at higher Reynolds numbers, the pressure gradient should not only overcome the viscous resistance but also the inertial force, which is proportional to the square of the velocity; thus, the momentum equation (taking a two-dimensional incompressible flow as an example) reads [14]

$$\rho\left(\frac{1}{\varphi}\frac{\partial\vec{V}}{\partial t} + \frac{1}{\varphi}\nabla \cdot \left(\frac{\vec{V}\vec{V}}{\varphi}\right)\right) = -\frac{1}{\varphi}\nabla(\varphi p) + \frac{\mu}{\varphi\rho}\nabla^2\vec{V} - \frac{\mu}{K}\vec{V} - \frac{\rho C_F}{\sqrt{K}}\left|\vec{V}\right|\vec{V} \tag{3}$$

with $C_F$ being the Forchheimer coefficient.

In Equation (3), the Darcy term $-\frac{\mu}{K}\vec{V}$ and the Forchheimer term $-\frac{\rho C_F}{\sqrt{K}}\left|\vec{V}\right|\vec{V}$ represent the effects of the viscous force and the inertial force in porous media, respectively.

Since the permeability $K$ is determined by the structural parameters of porous material and does not change linearly with the structural parameters, together with the Forchheimer coefficient $C_F$, they can be calculated by the empirical Equations (4) and (5):

$$K = \frac{\varphi^3 d_p^2}{150(1-\varphi)^2} \tag{4}$$

$$C_F = \frac{1.75}{\sqrt{150}\varphi^{3/2}} \tag{5}$$

where $d_p$ is the cell diameter of the porous material.

### 2.2. Aeroacoustics Governing Equations: FW-H Acoustic Analogy

Understanding airfoil trailing edge noise mechanisms requires a mature numerical tool. Computational aeroacoustic (CAA) methods for the noise generation of airfoil were widely applied, benefiting from the development of high-performance computing technology. However, three-dimensional noise modeling using CAA/LES is still a heavy task. A hybrid numerical method has been widely used in noise prediction to save computing costs, in which the flow field and acoustic field can be solved separately. In order to investigate the radiation characteristics of flow-induced noise, Ffowcs Williams and Hawkings [29] (FW-H) developed the original formula based on Lighthill's acoustic analogy by rearranging the continuity equation and momentum equation into a non-homogeneous wave equation with sound sources. Lighthill's equation is initially derived from the mass and momentum equations of compressible NS equations, which is given by Equation (6).

$$\frac{\partial \rho}{\partial t} + \frac{\partial (\rho u_j)}{\partial x_j} = 0, \quad \frac{\partial (\rho u_i)}{\partial t} + \frac{\partial (\rho u_i u_j)}{\partial x_j} - \frac{\partial p_{ij}}{\partial x_j} = 0 \tag{6}$$

When applying the Heaviside function $H(f) = \begin{cases} 1 & f > 0 \\ 0 & f < 0 \end{cases}$ and describing the generalized variables, the density, momentum, and compressive tensors can be written as

$$\begin{cases} \widetilde{\rho} = \rho' H(f) + \rho_0 \\ \widetilde{\rho u_i} = \rho u_i H(f) \\ \widetilde{p}_{ij} = p'_{ij} H(f) + p_0 \delta_{ij} \end{cases} \tag{7}$$

where variables with the subscript '$_0$' represent the values in undisturbed medium, the primed values of $\rho$ and $p_{ij}$ represent the difference between the values in the real condition and those in the undisturbed medium, and $f$ indicates the control surface shape and its motion, which can be defined as $f(x, t)$. Applying the generalized functions to Equation (6) yields

$$\begin{array}{l} \frac{\partial}{\partial t}[\rho' H(f)] + \frac{\partial}{\partial x_i}[\rho u_i H(f)] = \rho_0 u_i \frac{\partial H}{\partial x_i} \\ \frac{\partial}{\partial t}[\rho u_i H(f)] + \frac{\partial}{\partial x_j}[(\rho u_i u_j + p_{ij})H(f)] = p_{ij} u_i \frac{\partial H}{\partial x_j} \end{array} \tag{8}$$

with $\delta(f) = \frac{\partial H(f)}{\partial f} = \begin{cases} \infty & f = 0 \\ 0 & f \neq 0 \end{cases}$, which is the Dirac delta function.

By combining the time-derivative terms of the continuity equation and the spatial derivative terms of the momentum equation, the differential FW-H equation is derived, where $c_0$ is the speed of sound, $p_0$ is the acoustic pressure, $T_{ij}$ is the Lighthill stress tensor, and $l_i = p'_{ij} n_j$ is the local force vector exerted on the control surface.

$$\frac{1}{c_0^2} \frac{\partial^2 p'}{\partial t^2} - \overline{\nabla}^2 p' = \frac{\partial}{\partial t}[\rho_0 v_n \delta(f)] - \frac{\partial}{\partial x_i}[l_i \delta(f)] + \frac{\overline{\partial}^2}{\partial x_i \partial x_j}[T_{ij} H(f)] \tag{9}$$

Here, $v_n$ is the normal velocity of the control surface.

In Equation (9), the first term on the right side indicates the monopole source caused by a surface acceleration or displacement; the second term determines the dipole source caused by surface pressure disturbances; the third term represents the quadruple sound source induced by turbulent flows. It is worth mentioning that rather than the built-in methods in Ansys Fluent, there are more advanced numerical methods to solve the viscous resistance and inertial resistance coefficients, such as the element-free Galerkin method [30] and the generalized 2D Bézier method [31].

### 2.3. Numerical Configurations

Flow and acoustic simulations were carried out for a NACA0018 airfoil, which has a chord length of *c* = 0.2 m and a span length of *l* = 0.05 m. While keeping its baseline geometry unchanged, the 20% chord length measured from the trailing edge was set as a porous domain. Two porous airfoils, named the fully porous type and the suction-side porous type, are depicted in Figure 1. The hypothesis and simplifications for porous material were made as follows: (1) the metal foam is assumed to be homogeneous and isotropic; (2) the flow in the porous region satisfies Darcy's law; (3) the physical parameters of porous materials are constant.

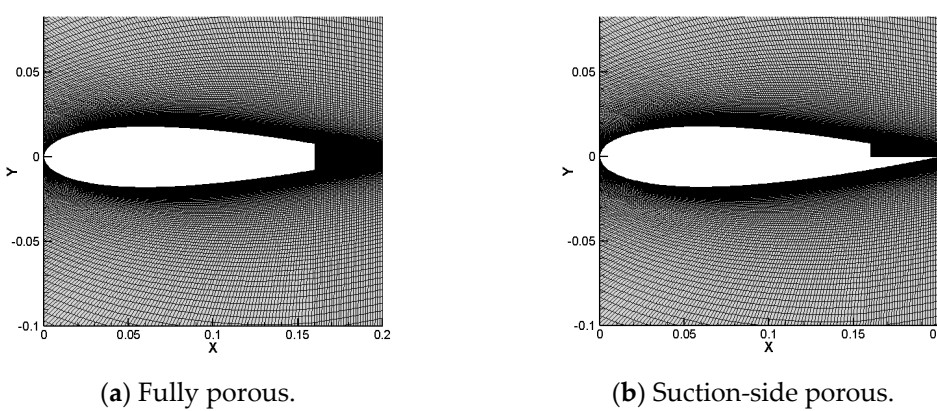

(**a**) Fully porous.          (**b**) Suction-side porous.

**Figure 1.** Computational mesh of porous airfoils.

The flow field was computed in a circular computational domain with a radius of 15*c* and a span of 0.25*c*. The whole computational domain was meshed with hexahedron elements in both the outflow field and the porous flow field. The overall cells for the solid airfoil was about $1.24 \times 10^6$. The outer boundary was set as a velocity inlet on the left side and a pressure outlet on the right side; the boundary interface between the solid part of the airfoil and the porous domain satisfied the no-slip wall condition. In order to interpolate and transfer the calculation data, the boundary between the porous domain and the outer fluid domain was taken as an interface, and both sides of the interface were paired by the matching method. The settings of the porous zone were derived from the physical parameters of aluminum foam, and all the parameters were obtained from Carpio et al. [23], which consisted of a cell diameter of 800 μm, a porosity of 0.917, and a permeability of $2.7 \times 10^{-9}$ $m^2$. Thus, the viscous resistance and inertial resistance coefficients could be calculated according to Equations (4) and (5). The numerical simulations were performed using the commercial software ANSYS® FLUENT 2019.

In order to study the near-field and far-field directivity of sound radiation, 12 receiver points were named A1–A12, and another group of 12 receiver points were named a1–a12. They were located at distances of 10*c* and 0.8*c* from the airfoil geometrical center, respectively. The receiver points were equally distributed in the azimuthal direction with an interval of 30° in each case, as shown in Figure 2. Additionally, for the convenience of comparing the sound pressure level with the experimental data [23], another receiver point B was set at 1.43 m right above the geometric center of the airfoil section.

The analysis of the flow and acoustics was based on the solutions performed with LES, where the Smagorinsky–Lilly eddy viscosity model was chosen as the sub-grid-scale model. The free-stream velocity was 20 m/s, with a chord-based Reynolds number of $2.63 \times 10^5$. The second-order upwind scheme was adopted in the spatial discretization of the convective term, and the SIMPLE algorithm was applied for the pressure-velocity coupling of the continuity and momentum equations. The considerations of the time scales for the LES were mainly divided into two aspects to meet the requirements of the flow field calculation and the acoustic calculation. The time step should be less than the value of the mesh characteristic size divided by the characteristic flow speed. Furthermore,

the maximum resolvable frequency was proportional to $1/\Delta t$, from which the desired frequency was estimated. The time step was set as $1 \times 10^{-5}$ s, while the transient flow characteristics were captured, and the acoustic simulation was activated after $4 \times 10^4$ steps when the flow entered into a quasi-steady state. For each acoustic calculation period, $2 \times 10^4$ steps were included, covering 0.2 s in total.

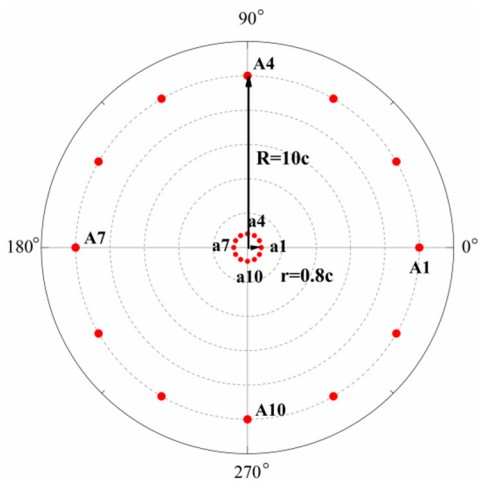

**Figure 2.** Acoustic receiver arrangement.

Before carrying out the flow and aero-acoustics simulations, the independence and reliability of the grid configurations were verified in order to ensure the accuracy of the numerical results and the computational efficiency. As listed in Table 1, five different grids for the baseline airfoil were generated to perform the grid independency test, all of which shared the same wall-normal height size of $\Delta y = 7.8 \times 10^{-5}$ m and wall-normal resolution of y+ = 5 in the immediate vicinity of the airfoil. The results displayed in Figure 3 show that the errors of the lift and drag coefficients of the latter four grids were controlled within 1% under angles of attack between 0° and 10° with an increment of 2°.

**Table 1.** Grid parameters (grid case 3 was adopted for further simulations).

| Grid Case | Chord-Wise Nodes | Span-Wise Nodes | Cells |
|---|---|---|---|
| 1 | 125 | 10 | $17.32 \times 10^4$ |
| 2 | 168 | 15 | $31.43 \times 10^4$ |
| **3** | 310 | 15 | $103.58 \times 10^4$ |
| 4 | 420 | 20 | $191.40 \times 10^4$ |
| 5 | 310 | 31 | $219.20 \times 10^4$ |

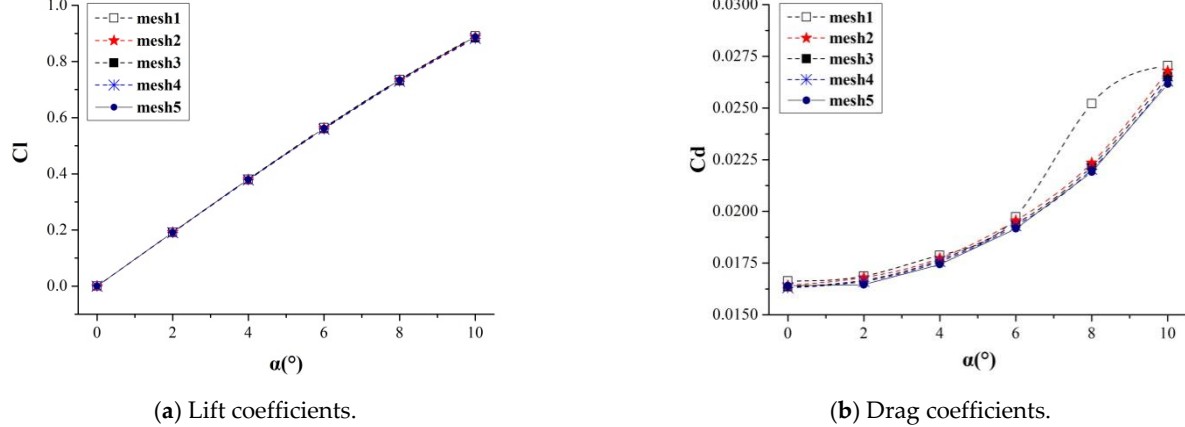

(**a**) Lift coefficients.

(**b**) Drag coefficients.

**Figure 3.** Lift and drag coefficients for independence verification.

Figure 4 shows the acoustic results of the two selected grids with span-wise distribution (i.e., grids 3 and 5) compared to those of the experiment in [23]. According to the sound spectra, the calculation results of the two grids are relatively close. In this paper, grid 3 was adopted to guarantee the computational efficiency and the computational accuracy.

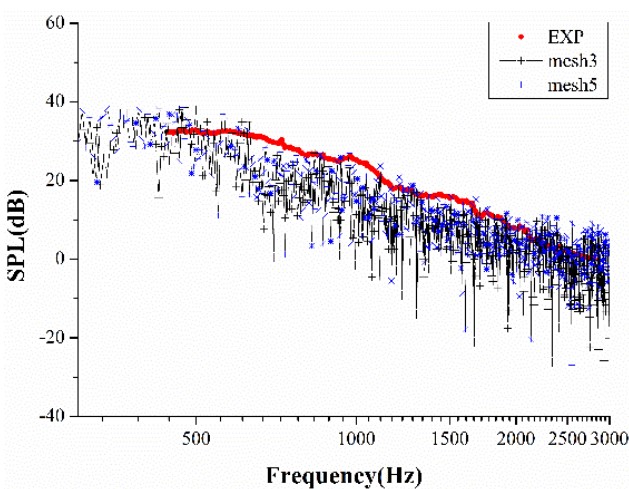

**Figure 4.** Acoustic simulations of grids 3 and 5 for a NACA0018 airfoil with a solid trailing edge.

The numerical results of the time-averaged surface pressure coefficient, which was extracted at the spanwise location $z = 0.025$ based on grid 3, were compared with the wind tunnel experimental data [32] at $\alpha = 6°$ and a Reynolds number of $1.6 \times 10^5$. Good agreement was achieved, as depicted in Figure 5.

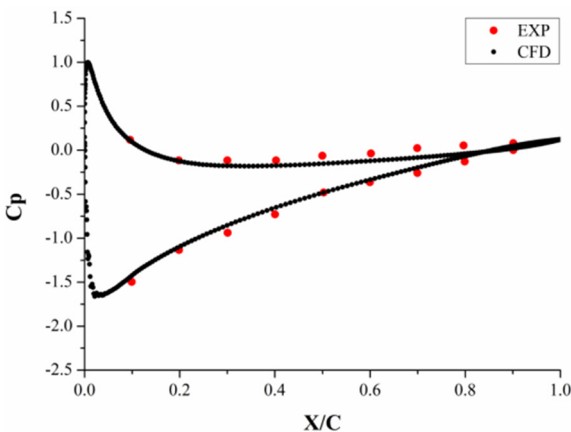

**Figure 5.** Distribution of pressure coefficients along the airfoil surface.

## 3. Results and Discussions

### 3.1. Flow Characteristics near Trailing Edge

Due to the presence of the porous material, the flow feature had an apparent change, as shown in Figure 6. The transient flows near the trailing edge of both the solid and porous airfoils are shown at an angle of attack of $\alpha = 0°$. Due to the permeability of the porous media, the flow passed through the porous region at the trailing edge and thus changed the surrounding flow pattern. The most notable phenomenon observed was that the air flowed in the porous region and formed recirculation structures inside. The velocity magnitude in the porous media was approximately 10% of the free stream wind speed under the given porous material. Due to the larger pressure gradient from the lower (solid) to the upper surfaces (porous) at the trailing edge, the flow on the pressure side circulated towards the suctions side that formed a bubble inside the porous region.

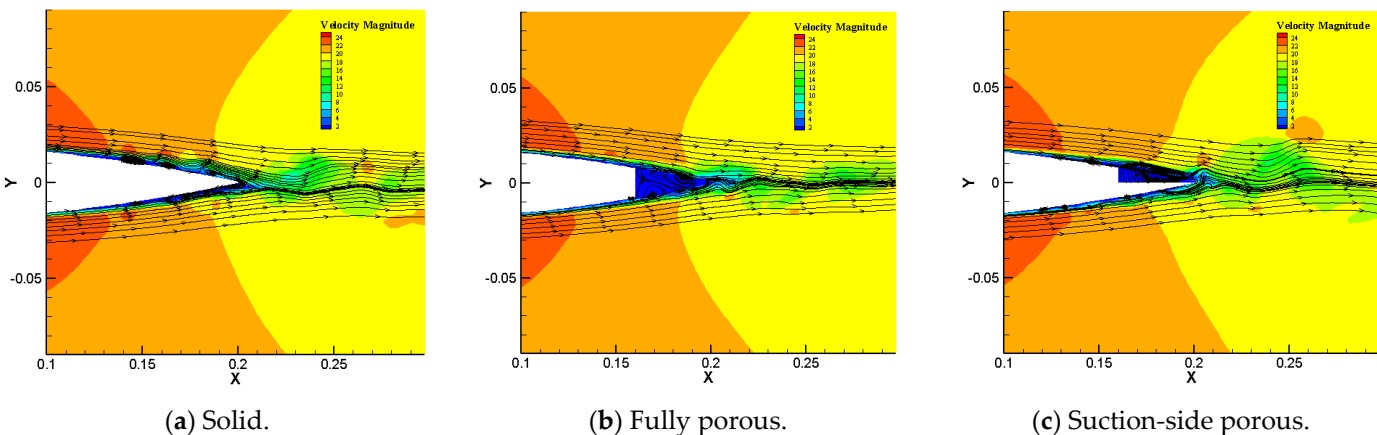

**Figure 6.** Transient flow with streamlines at trailing edge ($\alpha = 0°$, t = 0.45 s).

Since the changes in the flow in the porous domain were more significant for the fully porous airfoil, the turbulent flows near the trailing edge under different angles of attack were analyzed. As illustrated in Figure 7, at four different angles of attack, $\alpha = 0°$, $\alpha = 2°$, $\alpha = 4°$ and $\alpha = 6°$, the recirculation of streamlines inside the porous domain were evident, while the suction side recirculation continuously grew when the angle of attack was larger than $0°$. With a further increase in the angle of attack, as shown in Figure 7c,d, it is clear that the recirculation structures started to disappear and the flow had a tendency to move from the pressure side to the suction side, as indicated by the streamlines, which was caused by the growing pressure difference near the trailing edge.

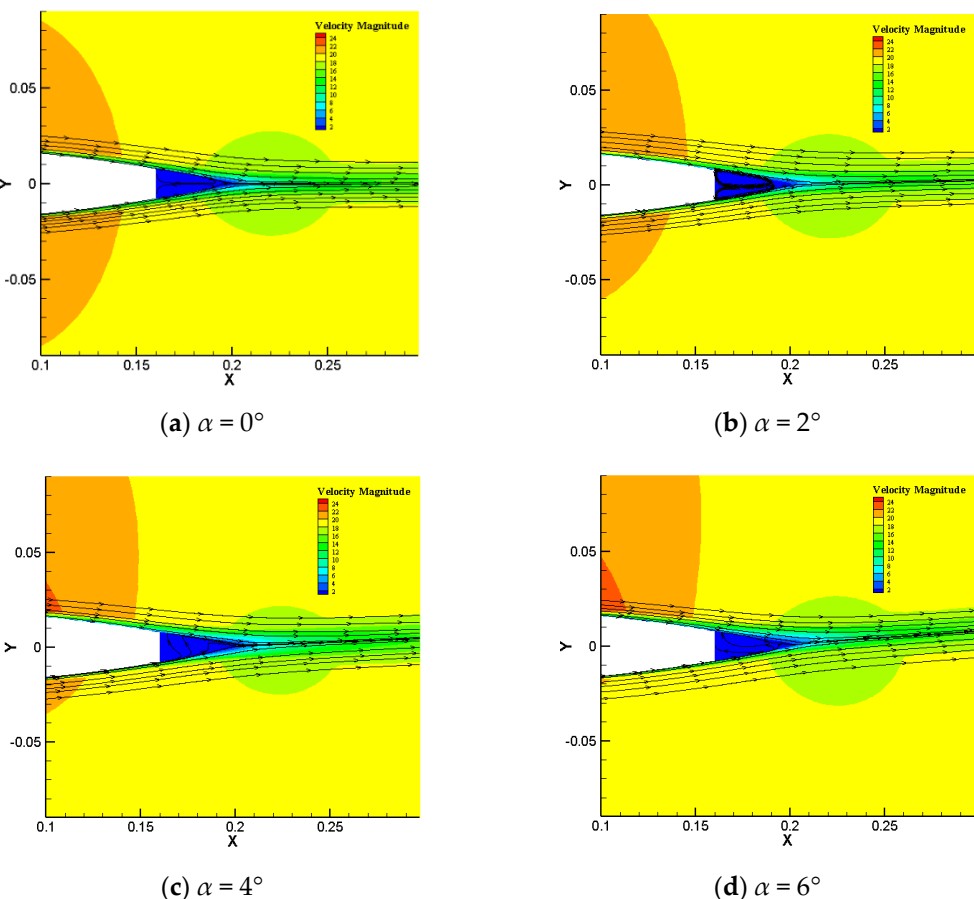

**Figure 7.** Steady-state streamlines near the trailing edge of a fully filled porous NACA0018 airfoil.

### 3.2. Pressure Distribution along the Airfoil Surface

The aerodynamic performance was analyzed from the distribution of the pressure coefficient along the airfoil surface at the angles of attack of $0°$, $2°$, $4°$, and $6°$. As depicted in Figure 8a, at an angle of attack of $0°$, the pressure difference between the suction and the pressure sides were obvious along the whole chord-wise direction for the suction-side porous airfoil because the symmetrical geometry of the NACA0018 was ruined due to the one-side porous trailing edge. Thus, at $\alpha = 0°$ for the suction-side-filled airfoil, a non-zero lift force could be obtained. In Figure 8b–d, it can be seen that the $C_p$ distribution for the fully porous trailing edge was similar to that of the solid airfoil. The suction-side-filled airfoil had a smaller area closed by the $C_p$ curve before $x/c = 0.8$ ($c$ is the chord length), which led to a smaller lift. However, starting from $x/c = 0.8$, where the porous medium started, the pressure values on the suction and pressure sides coincided with each other for the fully filled airfoil, resulting in a loss in the total lift force. Meanwhile, the suction-side-filled airfoil still had a pressure difference, which contributed to a larger lift and could offset its $C_p$ loss before $x/c = 0.8$. From the integrated lift coefficients listed in Table 2, it is obvious that the lift force created by the solid airfoil was the largest, while the lift obtained from the suction-side-filled airfoil was larger than that from the fully filled airfoil.

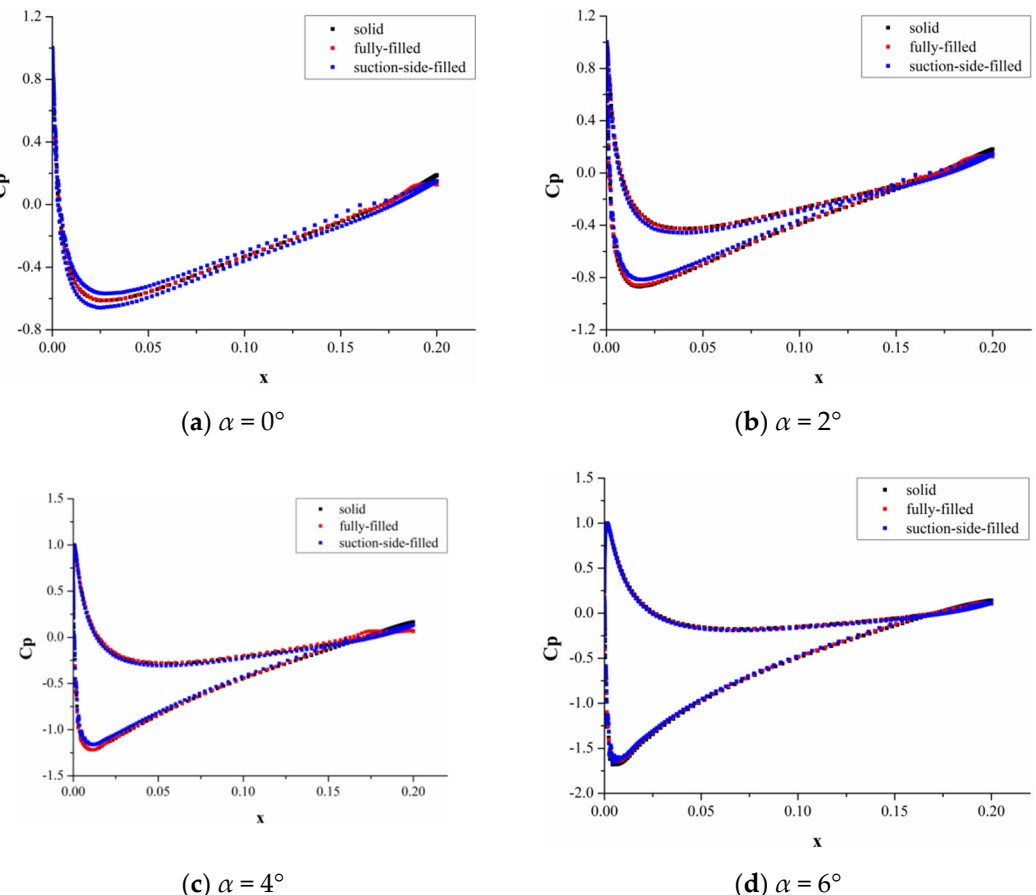

**Figure 8.** Pressure coefficient along the airfoil surface at different angles of attack.

Finally, the aerodynamic performances of the three airfoils are demonstrated in Figure 9. In general, both the two porous airfoils show a reduced aerodynamic performance compared to the solid one, which is mainly marked by a decreased lift coefficient and an increased drag coefficient. For the fully porous trailing edge airfoil, the flows on the upper and lower sides interacted with each other, and, therefore, the pressure difference was mainly eliminated after $x = 0.8c$ at different angles of attack. For the suction-side porous

trailing edge airfoil, a non-zero lift was seen at $\alpha = 0°$ and its distribution of $C_p$ along the airfoil surface was closer to that of the solid airfoil as the angle of attack grew.

**Table 2.** Lift coefficient comparison at different angles of attack for the solid, fully filled, and suction-side-filled airfoils.

| $\alpha^o$ | Solid | Fully Filled | Suction-Side-Filled |
|---|---|---|---|
| 0 | $-4.55 \times 10^{-6}$ | $4.61 \times 10^{-5}$ | 0.17253 |
| 2 | 0.20126 | 0.19894 | 0.19989 |
| 4 | 0.38778 | 0.38142 | 0.38845 |
| 6 | 0.56762 | 0.55526 | 0.56973 |
| 8 | 0.74270 | 0.73436 | 0.73925 |
| 10 | 0.88547 | 0.86027 | 0.87523 |

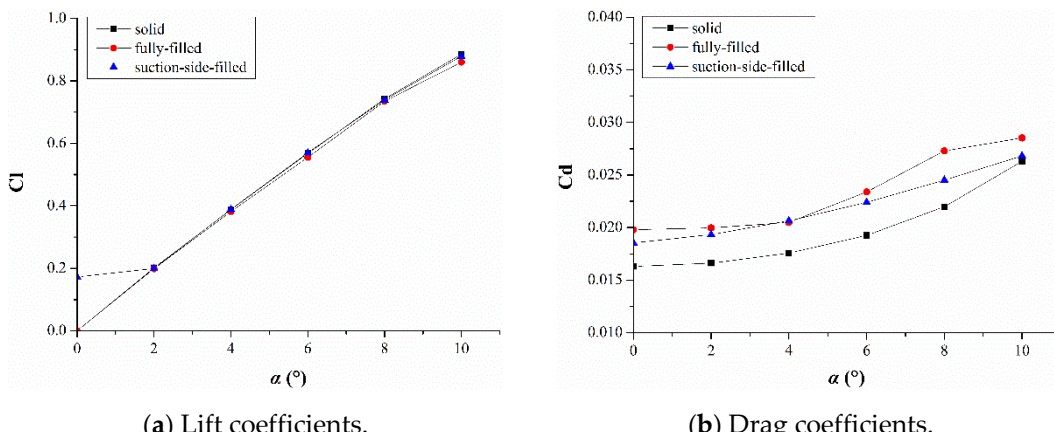

(**a**) Lift coefficients.　　　　　　(**b**) Drag coefficients.

**Figure 9.** Comparison of lift and drag performances.

### 3.3. Boundary Layer Velocity and Thickness Distributions

Figure 10 shows the horizontal velocity component along the wall normal directions from the positions of $x = 0.5c–1c$ at $\alpha = 0°$. For the porous airfoils, the velocity reduction appears along the whole chord-wise direction, which reflects the fact that the flow resistance exerted by the porous material consumed a part of the kinetic energy and therefore slowed down the flow speed nearby. It is also seen in Figure 10a–c that the influence of the porous configuration was not significant at the upstream positions of $x = 0.5c$, $0.6c$ and $0.7c$. However, starting from $x = 0.8c$, where the porous material presents, the velocity distributions of the fully porous and suction-side porous cases gradually differ from each other. Due to the effect of the increased surface roughness at the trailing edge, the overall velocity in the wall boundary layer continuously decreased. In particular, for the suction-side porous airfoil, as described in Figure 10h, the normalized velocity of the suction-side filled airfoil at a wall distance of $y/c = 0.02$ was around 60% of that of the baseline airfoil.

Figures 11 and 12 illustrate the boundary layer thickness and momentum thickness at different stream-wise positions on the suction side at $x = 0.5c–1c$ at different angles of attack. The results explicitly show the remarkable differences at low angles of attack. Interestingly, the influence of porous material diminished at $\alpha = 4°$, when an apparent flow separation took place. As shown in Figure 11, at $\alpha = 0°$, the boundary layer thickness of the porous configurations was a few times larger than of the baseline case, where at $\alpha = 4°$, the boundary layer thickness distributions were fully overlapped with each other.

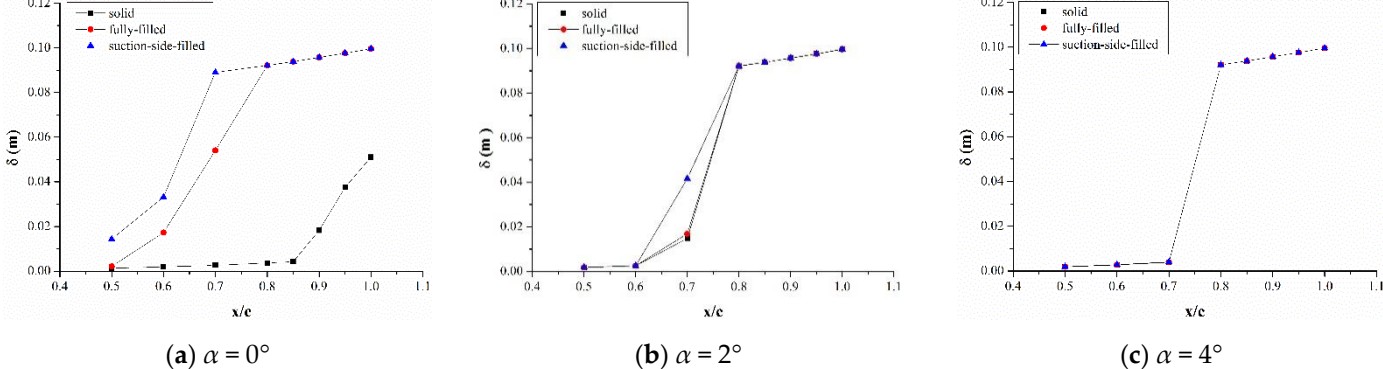

|  |  |  |  |
| :-: | :-: | :-: | :-: |
| (**a**) *x* = 0.5*c* | (**b**) *x* = 0.6*c* | (**c**) *x* = 0.7*c* | (**d**) *x* = 0.8*c* |
| (**e**) *x* = 0.85*c* | (**f**) *x* = 0.9*c* | (**g**) *x* = 0.95*c* | (**h**) *x* = 1.0*c* |

**Figure 10.** Evolution of horizontal velocity at different stream-wise positions on the suction side ($\alpha = 0°$).

|  |  |  |
| :-: | :-: | :-: |
| (**a**) $\alpha = 0°$ | (**b**) $\alpha = 2°$ | (**c**) $\alpha = 4°$ |

**Figure 11.** Boundary layer thickness at different stream-wise positions on the suction side airfoil surface.

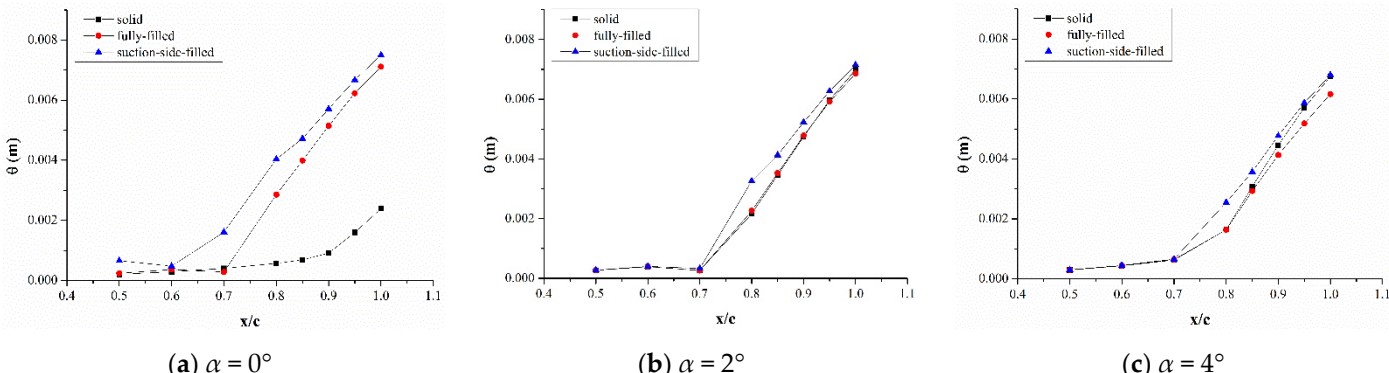

**Figure 12.** Boundary layer momentum thickness at different stream-wise positions on the suction-side airfoil surface.

The boundary layer momentum thickness distribution is somewhat more complicated as it is a function of the normalized velocity, which is integrated overall in the boundary layer. Along the chord-wise direction, the suction-side porous configuration had the largest momentum thickness compared to the other two cases, as shown in Figure 12. At a low angle of attack of $\alpha = 0°$, the porous trailing edge changed the developments of the boundary layer along the stream-wise positions, mainly shown as the prominent increase in the boundary layer thickness. As the angle of attack increased to $2°$, a light flow separation started at the position of $x = 0.7c$. It is seen that the fully porous configuration nearly had the same momentum thickness distribution as the baseline airfoil, which was similar to the thickness distribution. At a higher angle of attack of $\alpha = 4°$, the boundary layer thickness and momentum thickness of the porous airfoils are somewhat in agreement with those of the solid airfoil at different chord-wise positions. The interference produced by the porous material was relieved at higher angles of attack since a strong flow separation occurred on the trailing edge of the solid airfoil.

*3.4. Aeroacoustics Analysis*

In this paper, the analysis on noise generation from the three airfoil configurations was mainly based on the FW-H acoustic analogy. When the flow characteristics around the airfoil surface are known, the acoustic pressure in the far field can be obtained by an integration of Equation (9) over the airfoil surface, after which it is possible to generate the noise spectra from temporal FW-H data via Fourier transform. For the three airfoil configurations, an example of the acoustic pressure history recorded at receiver B during a time period of 0.4–0.45 s is given in Figure 13. It can be seen that at $\alpha = 0°$, the porous airfoils had the ability to suppress the sound pressure fluctuations considering the amplitude, which can directly reflect that porous airfoils do work for noise reduction, and as expected, the fully porous type obviously performed better, with a large reduction in the pressure amplitude. According to acoustic analogy, loading noise and thickness noise are the two dominant mechanisms for wind turbine blades; the latter considers the blade rotational effect, which does not apply in the current case. Figure 14 illustrates the power spectra of the lift fluctuations as they reflect the loading noise of an airfoil. According to the spectrum of the aerodynamic loading, the solid airfoil had broadband behavior with a considerably large magnitude of power spectral density in the frequency range of 100–1000 Hz. On the contrary, the suction-side porous airfoil exhibited smaller overall magnitudes and was limited to a rather narrow frequency range. As for the fully porous airfoil, the amplitudes were almost suppressed in a broad frequency band, replaced by a distinct dominant frequency with a much smaller peak value. To conclude, the power spectral density distributions of the two porous airfoils were relatively concentrated, and the discrete characteristics are more obvious. The energy density rates of the porous airfoils were attenuated at a wide frequency range.

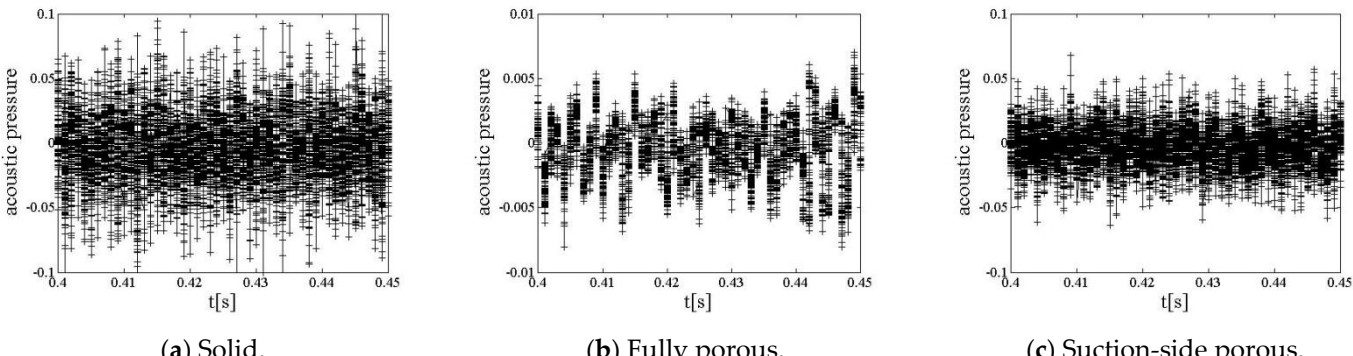

(**a**) Solid.  (**b**) Fully porous.  (**c**) Suction-side porous.

**Figure 13.** Acoustic pressure history ($\alpha = 0°$, t = 0.4–0.45 s).

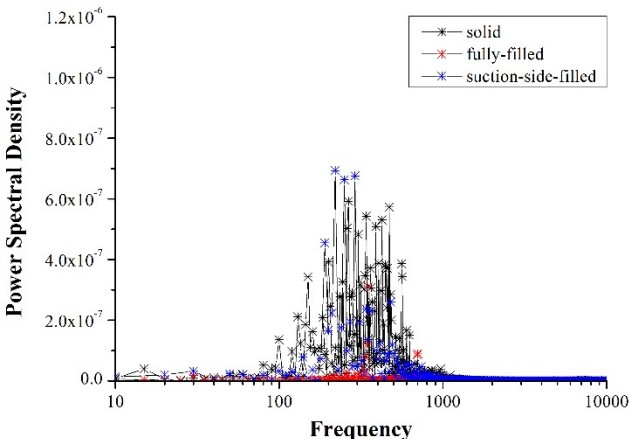

**Figure 14.** Power spectral density of lift force.

In Figure 15, the distribution of the sound pressure level (SPL) at receiver point B with $\alpha = 0°$ is shown in a frequency range of 0–3 kHz. At lower frequencies, the suction-side porous airfoil shows an evident SPL reduction compared to the solid airfoil, while the reduction was much more remarkable for the fully porous airfoil. However, for frequencies larger than 2.5 kHz, the SPL of the porous airfoils tended to exceed the reference solid airfoil. On the other hand, the SPL reduction in the mid and low frequencies contributed more to the overall sound pressure levels (OASPLs). In general, the two porous airfoils can realize noise reductions at different levels. For the solid airfoil, suction-side porous airfoil, and fully porous airfoil, the OASPLs were 69.2 dB, 64.9 dB, and 61.6 dB, respectively. Apparently, the noise reduction performance was more distinct for the fully porous airfoil due to the fact that the larger porous domain caused a greater loss of kinetic energy and led to smaller flow disturbances near the trailing edge. In addition, the suction and pressure sides of the fully porous airfoil were thoroughly communicated and the pressure fluctuation was reduced to a greater extent, thus achieving a superior noise reduction. However, as shown in the previous section, a trade-off between the aerodynamic performance and the noise reduction was achieved.

The vortex shedding frequency typically represents the peak frequency of the time-varying lift acting on the airfoil surface, which measures the periodic change of flow over the airfoil trailing edge. Since the power spectral density distribution is closely related to the sound pressure signal and its peak value can often be used to identify the vortex shedding frequency at trailing edge, Figures 14 and 15 can be analyzed together. First of all, the solid airfoil had a considerably high power spectral density in the frequency range of 100–1000 Hz. The two peaks of its power spectral density of the lift are located at 250 Hz and 461 Hz, corresponding to the two peaks of the SPL appearing at 265 Hz and 470 Hz, as reflected in Figure 15a. Additionally, the discrete behavior of the fully porous trailing edge airfoil is also observed in Figure 15b. The SPL peak is located at 353 Hz, which agrees well

with its vortex shedding frequency of 350 Hz. Similarly, the SPL peak of the suction-side porous airfoil is located at 246 Hz, and its vortex shedding frequency is about 250 Hz. Therefore, at a low angle of attack, the vortex shedding frequencies of these airfoils are very consistent with the sound pressure level peaks, and the effect of trailing edge vortex shedding on noise generation is obvious.

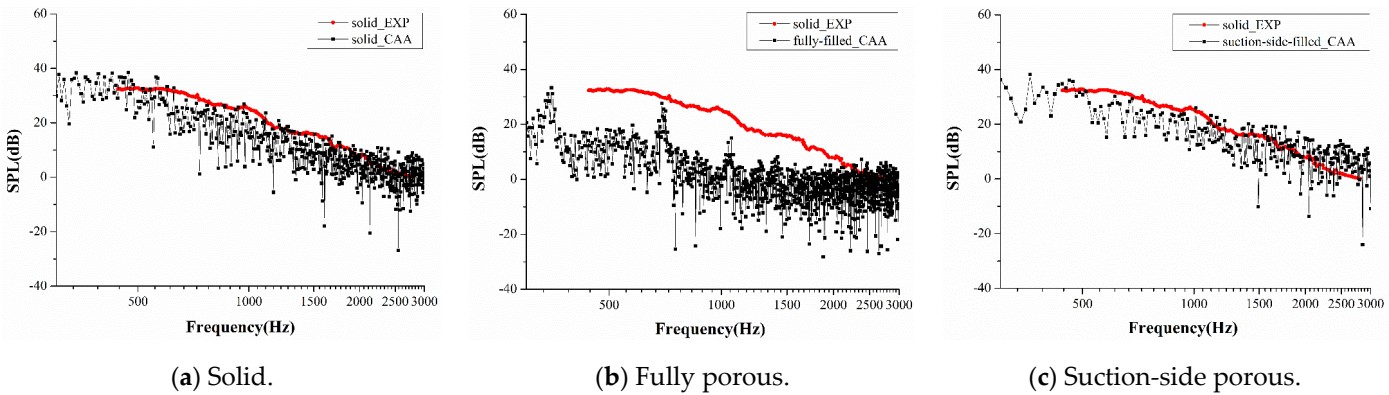

(**a**) Solid.      (**b**) Fully porous.      (**c**) Suction-side porous.

**Figure 15.** Sound pressure level of pressure signal for receiver B ($\alpha = 0°$).

Figure 16 shows the distribution of the sound pressure level for the suction-side porous airfoil under 8000 Hz at the angles of attack of 0° and 2°. As Figure 16a shows, the spectrum distribution of the SPL at $\alpha = 2°$ was similar to that at $\alpha = 0°$, while only the magnitude changed. At $\alpha = 2°$, the overall sound pressure levels were 58.96 dB and 54.89 dB for the solid airfoil and the suction-sided porous airfoil, respectively, with a reduction of 4.07 dB, which is slightly inferior to the case of $\alpha = 0°$. In addition, when focusing on the frequency range of 0–1000 Hz, as illustrated in Figure 16b, higher fluctuations were manifested at $\alpha = 0°$, while the spectrum is flatter at $\alpha = 2°$. This primarily confirms that the distinct tonal noise will gradually be smeared out with the increase in the angle of attack, leaving a broadband feature of the spectrum more distinguished.

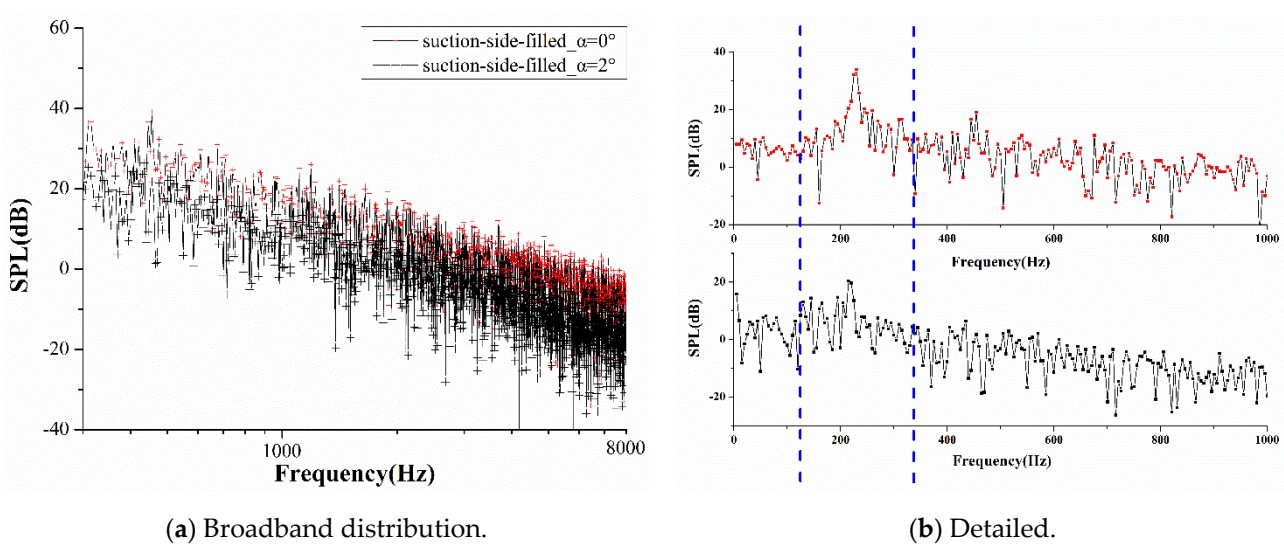

(**a**) Broadband distribution.      (**b**) Detailed.

**Figure 16.** Sound pressure level for the suction-side porous trailing edge airfoil at $\alpha = 0°$ and $\alpha = 2°$.

To distinguish the sound directivity of porous airfoils, several receivers around the airfoils were set to monitor the sound pressure levels. SPL signals from these receivers are shown in Figure 17, where Figure 17a presents the sound directivity pattern of the near field ($r = 0.8c$), and Figure 17b presents the far-field sound directivity pattern ($r = 10c$). Firstly, for the SPL distribution of both the near field and far field, the solid airfoil shows more distinct

low values along the leading edge and trailing edge, while exhibiting a significant SPL in the airfoil wall normal directions, which finally leads to a sound directivity resembling that of a compact dipole. As illustrated in Figure 17a, the SPL of the suction-side porous airfoil increased dramatically in the directions along the leading edge and trailing edge, leaving the depressions on the right side diminished. The whole directivity pattern resembles a distorted dipole, which is tilted towards the leading edge on the left side and more like a semicircle on the right side. As for the fully filled airfoil, the magnitudes in all directions were decreased as well, except along the leading edge and trailing edge directions, in which the SPL values along the directions of 0–30°, 150–210°, and 330–360° were largely spread out.

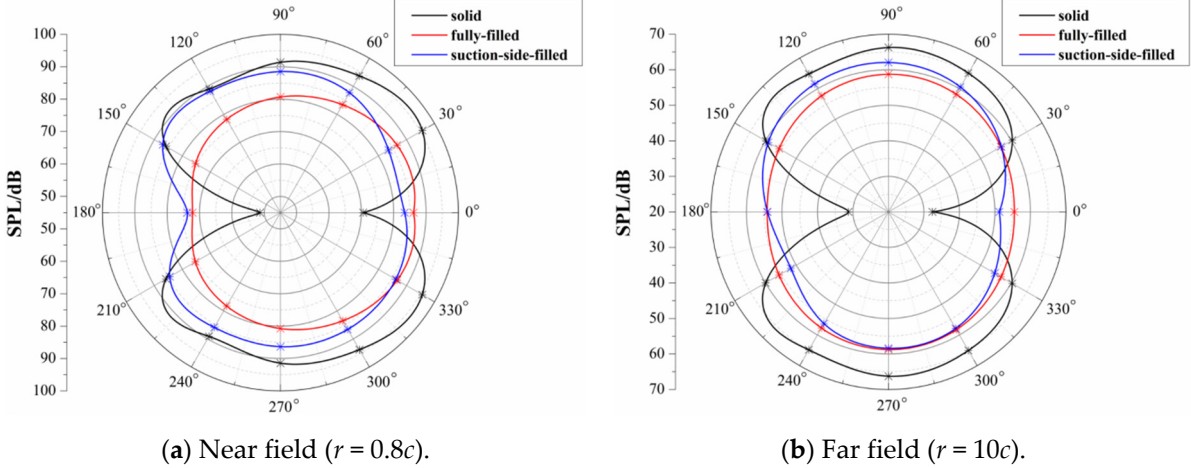

(**a**) Near field ($r = 0.8c$).　　　　　　(**b**) Far field ($r = 10c$).

**Figure 17.** Directivity patterns of OASPL ($\alpha = 0°$).

On the other hand, both of the two porous airfoils claimed excellent abilities for reducing noise in the directions of 15–150° and 210–345°, as shown in Figure 17a. Comparing to the solid airfoil, the fully porous configuration realized a quite impressive reduction within all the above-mentioned azimuth angles, while the suction-side porous airfoil only had a better performance near the trailing edge on the suction-side surface, especially for the angles of 15–30° and 330–345°, and it even made a higher reduction than the fully porous airfoil. With regard to the far-field noise, the effective azimuth angle for porous airfoils was reduced at 30–150° and 210–330°. In addition, a noise reduction ability is also displayed for the fully porous airfoil on the suction side, but it shows a similar effect on the pressure side, which indicates that the suction-side flow field may have reacted more strongly to the porous trailing edge. In general, the porous trailing edge achieved a considerable noise reduction in the large angular range.

## 4. Conclusions

In this paper, airfoil trailing edge noise reduction with a new filling method is proposed. A porous medium was only used on the suction side of trailing edge, given that the suction side filling method was less sensitive to the wall surface pressure and flow changes, whereas the fully filled trailing edge case led to a large aerodynamic loss. Based on computational fluid dynamics and computational aeroacoustic simulations, the flow and noise characteristics of the porous airfoils were investigated and compared with the baseline NACA0018 solid airfoil.

As expected, the flow could get through the trailing edge due to the permeability of the porous media. Flow recirculation occurred in the porous domain at low angles of attack, while as the angle of attack grew, the recirculation disappeared and the streamlines went directly through the pressure side towards the suction side. In such a case, the pressure difference between the pressure side and the suction side at the trailing edge became small, which resulted in a relatively low lift for the fully porous case. In general, fluctuations in

the time-series pressure coefficient for the two porous airfoils were weakened and their $C_p$ distributions were very consistent with that of the solid airfoil except at the trailing edge. It was also found that both of the two porous airfoils showed a suppression of the boundary layer velocity after $x = 0.8c$ when $\alpha = 0°$. The boundary layer thickness and the momentum thickness had a dramatic increase starting from the chord-wise position at $x = 0.5c$ towards the trailing edge.

It was shown that the new filling method performed well regarding noise reduction and aerodynamic performance. The sound pressure of the porous airfoils was suppressed with respect to its amplitude of fluctuation, and the power spectrum density showed a rather narrowed down frequency range compared to that of the solid airfoil, which indicates that the suction-side porous type could slightly reduce the SPL at low frequencies. The result is that a noise reduction of 4.3 dB was achieved at $\alpha = 0°$. A similar performance was achieved at higher angles of attack. Taking $\alpha = 2°$ as an example, a slightly decreased value of 4.07 dB was obtained. As for the sound directivity, the proposed porous airfoil also claimed good abilities of reducing noise in a large range of direction angles.

Still, the present work has some limitations, such as the lack of research on different porous materials, which plays a decisive role in flow and acoustic characteristics. As for future investigations, more emphasis will be placed on the materials and bolder and more innovative layouts of porous materials. The research on porous airfoils has been focused on reducing the trailing edge turbulent boundary layer flow noise, that is, attached flow or flow with limited separation at the trailing edge. In this sense, a porous medium is only considered as a passive device to reduce noise in normal wind turbine operation conditions. Fortunately, large flow separations need to be avoided for any modern wind turbine design and operation, and the turbulent boundary layer trailing edge noise is known as the major wind turbine noise source, which attracts extensive studies. On the other hand, there are few works considering airfoil noise under large flow separations, especially if such noise becomes periodic, which might be annoying for the nearby residents, even at a relatively low SPL. Therefore, more works should be carried out for controlling stall noise or noise under large separations.

**Author Contributions:** Methodology, W.Z.; Software, J.C.; Investigation, W.S.; Resources, Z.S.; Data curation, J.L.; Writing—original draft, W.Z.; Writing—review & editing, W.S.; Visualization, G.G.; Supervision, W.Z.; Project administration, W.Z. All authors have read and agreed to the published version of the manuscript.

**Funding:** This research was funded by Ministry of Science and Technology, 2019YFE0192600, and the National Nature Science Foundation under grant numbers 51905469 and 11672261.

**Institutional Review Board Statement:** Not applicable.

**Informed Consent Statement:** Not applicable.

**Data Availability Statement:** Not applicable.

**Conflicts of Interest:** The authors declare no conflict of interest.

## Nomenclature

| | |
|---|---|
| $\vec{V}$ | Darcy velocity (m/s) |
| $\vec{v}$ | intrinsic averaged velocity (m/s) |
| $\varphi$ | porosity of material (-) |
| $K$ | permeability (-) |
| $C_F$ | Forchheimer coefficient (-) |
| $\rho$ | density (kg/m$^3$) |
| $\mu$ | dynamic viscosity (N·s/m$^2$) |

| | |
|---|---|
| $H(f)$ | Heaviside function (-) |
| $f(x, t)$ | control surface shape and its motion (-) |
| $\delta(f)$ | Dirac delta function (-) |
| $c_0$ | speed of sound (m/s) |
| $p_0$ | acoustic pressure (Pa) |
| $d_p$ | cell diameter of porous material (-) |
| $\delta$ | boundary layer thickness, when the local velocity is 0.99 of the edge velocity (m) |
| $\delta^*$ | boundary layer displacement thickness, $\delta^* = \int_0^\delta \left(1 - \frac{u}{u_e}\right) dy$ (-) |
| $\theta$ | boundary layer momentum thickness, $\theta = \int_0^\delta \frac{u}{u_e} \left(1 - \frac{u}{u_e}\right) dy$ (m) |
| $u_e$ | edge velocity (m/s) |

## Appendix A

There is no need to add any user defined code in ANSYS since the porous zone can be directly defined in the 'cell zone conditions' module by giving inputs such as the inertial and viscous terms. A porous zone is modeled as a special type of fluid zone in ANSYS software. To indicate that the fluid zone belongs to a porous region, enable the 'Porous Zone' option in the 'Fluid dialog box' and specify the parametric settings.

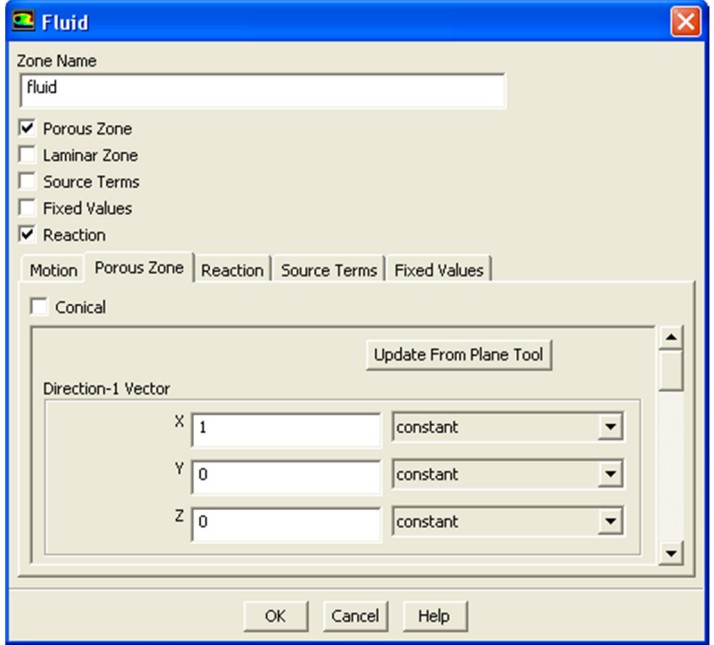

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
