# Peer review of "Numerical Study on Flow and Noise Characteristics of an NACA0018 Airfoil with a Porous Trailing Edge"

_sustainability, doi:10.3390/su15010275_

Round 1

Reviewer 1 Report

Based on the following comments and some serious observations about the methodology and analyses, I do not recommend the publication of this manuscript in Sustainability. I have 

Why did authors choose to use LES for this work? Because the simulations are 3D and transient in nature, how were the mesh sizes selected? Should there be not any justification for choosing length and time scales for LES without going into extensive grid-independence study. Considering the Reynolds number here, how does the mesh size of order 10^6 seem suitable for LES?

Because the authors have used wind turbine as the motivation for this study, how is the wind speed of 20 m/s justified here?

What is the reference for the experimental results in Fig. 4?

At what spanwise location on the wing, was Cp computed for the data in Fig. 5? Is it instantaneous Cp or averaged profile?

In Fig. 6c, the bubble is formed by the stepped airfoil. How does the porosity play a role here?

This study would have been more relevant if the aerodynamic and aeroacoustic performance be computed around the stall angle. It is because the wind turbine blades undergo dynamic stall during their rotation. 

Different scales on the y-axis have been used for the three plots in Fig. 13. I strongly suggest that the plots should be on the made on the same scales for a clear and more precise comparison.

At alpha = 0 deg, the lift should be zero for the solid airfoil. There should be no oscillation since the shear layers would remain attached with the body. The very low levels of spectral density for the lift force in Fig. 14 seems numerical and not physical. This observation shows that the analyses in this manuscript should be more rigorous and accurate.

Author Response

Dear Editor and reviewers,

We would like to thank you for your encouragements concerning our paper entitled “Numerical study on flow and noise characteristics of a NACA0018 airfoil with a porous trailing edge” (Ms. Ref. No.:  sustainability-1922874), submitted to Sustainability. Your comments are very helpful for improving the quality of our paper. The authors have revised our manuscript carefully in accordance with your valuable comments. According to the journal requirement, we submitted both an original marked-up copy using ‘track changes’, and a clean version of the revised manuscript.

- In our responses, the mentioned equations, figures, references numbers are referred to the update new version, unless specified separately;

- The author’s responses are in boldface.

Below we would like to address all the comments raised by Editor and reviewers.

Best Regards,

All authors

Reviewer 2 Report

Review report for “Numerical study on flow and noise characteristics of a NACA0018 airfoil with a porous trailing edge”, Sustainability-1922874

This paper researches the aerodynamic characteristic and acoustic characteristic of NACA0018 airfoil with a porous trailing edge. Some points of this paper need revise:

Introduction

1.     The first paragraph of introduction needs some revision. It is important to give the research findings of the references with proper comments rather than just report the work of other researchers.

2.     In the introduction, the references are not properly classified. Besides, what is the current state of the research direction? Although many references are cited, the references are not properly commented.

3.     Besides, the introduction part needs to become more concise while provide sufficient background knowledge with proper comments.

Computational Methodology

4.     The governing flow equations: modified Navier-Stokes equations and the governing aeroacoustic equations: FW-H acoustic analogy are introduced with reason given. However, how to use the commercial software Ansys Fluent to achieve these is not given in the paper. Please add the specific method to achieve the simulation calculation of modified Navier-Stokes equations and FW-H acoustic analogy in the commercial software Ansys Fluent.

Results and Discussions

5.     The results and discussion needs revise for concise while maintaining the useful research findings. Besides, the analysis needs more depth.

6.     Please give the definition of vortex shedding frequency.

7.     Figure 13, the vertical axis scale is suggested to be same of the three types of airfoil.

Additional comments:

8.     Please add a nomenclature after the conclusion and provide the calculation formula of some variable, for example, the formula of boundary layer thickness and momentum thickness.

9.     The format of the reference [30] needs amended.

Author Response

(The authors gave the same response as above.)

Reviewer 3 Report

Please read and “fully” address the comments listed below:

  1. The ABSTRACT is not written in a logical order. Start with an overview of the topic and a rationale for your paper. Describe the methodology you used and the general outline of the manuscript. Also, in the end, state the result in more detail (i.e., provide some numbers).

  2. Please fully introduce all the variables used in the equations (e.g., eq.8). Did you select all the equations (3-9) from a certain reference? Or did you derive some of them? Please clarify.

  3. Considering ANSYS® FLUENT 2019 software you used for numerical analysis, did you do any sensitivity analysis to determine the optimum mesh size?

  4. In Fig. 3, please justify the small discrepancy between the experimental and numerical results.

  5. Fig 14 (power spectral density of lift force) needs to be more explained in your manuscript.

  6. Conclusion: Can authors highlight future research directions and recommendations? Also, highlight the assumptions and limitations (e.g 1-2 shortcoming(s) of the present study)? Besides, recheck your manuscript and polish it for grammatical mistakes (you can use “Grammarly to quickly edit your document).

  7. In your manuscript, the viscous resistance and inertial resistance coefficients were numerically estimated using NSYS® FLUENT 2019 software. However, you can use strong and novel "Galerkin" and "Bezier" numerical methods to solve your problem. For this reason, please reference the following two papers, in which the method described in each ("Galerkin" and "Bezier") can be leveraged to numerically solve for viscous resistance and inertial resistance coefficients

“Galerkin Method”:

    • Zhang, J. P., Wang, S. S., Gong, S. G., Zuo, Q. S., & Hu, H. Y. (2019). Thermo-mechanical coupling analysis of the orthotropic structures by using element-free Galerkin method. Engineering Analysis with Boundary Elements, 101, 198-213.

    “Bezier Method“:

    • Kabir, H., & Aghdam, M. M. (2021). A generalized 2D Bézier-based solution for stress analysis of notched epoxy resin plates reinforced with graphene nanoplatelets. Thin-Walled Structures, 169, 108484. 

Author Response

(The authors gave the same response as above.)

Round 2

Reviewer 1 Report

I thank the reviewers to consider my suggestions for their revised manuscript. However, my important and major comments have not been incorporated in the revised document. I sincerely believe that the manuscript in its current form should not be recommended for publication in Sustainability. 

Author Response

Reviewer #1

Thank you very much for reviewing our manuscript. We really appreciate your good comments and suggestions that have significantly improved our manuscript. We have carefully considered all your comments and suggestions in the revised version. Please kindly find the response to all of your comments:

Reviewer:

I thank the reviewers to consider my suggestions for their revised manuscript. However, my important and major comments have not been incorporated in the revised document. I sincerely believe that the manuscript in its current form should not be recommended for publication in Sustainability. 

Re: Thank you for your comment, we fully understand the reviewer’s concerns. According to the previous comments, the major concerns from the reviewer are: (1) the use of LES for the 3D simulations and its grid sensitivity study; (2) flow around stall angle of attack; (3) the lift should be zero at alpha=0.

The authors wish to take this chance to further elaborate these comments. For the comment (1) It is not the only choice to use LES, but LES is the most nature choice, DNS is computationally too expensive for turbulence flow simulations, RANS cannot provide time dependent fluctuations from the flowfield and thus cannot be used for acoustic simulations. (some explanations are added in section 2. (2) There are totally 5 grid topologies presented in the paper which are also compared with experiments to proof the accuracy as well as computational efficiency. (2) Flow at stall angle of attack is to be controlled for modern wind turbines, only for stall regulated wind turbines the stall separation noise mechanism will become the dominant noise source. Turbulent boundary layer trailing edge noise is the dominant noise source for today’s wind turbine (pitch regulated and variable speed control) which happens at low angle of attack. We added some explanations at the beginning of the introduction. (3) the lift is zero if we take the mean value of the time history data, but for unsteady simulation, the fluctuation around mean value is what we desired to see, that is the input for aerodynamic noise generation. If the lift fluctuation is zero, there will be no aerodynamic noise generation because there is no flow perturbation.

More detailed changes are marked in the ‘Track Change Version’ of the manuscript.

Reviewer 2 Report

The revised paper has been reviewed. But the improvement is very limited. There still exists some errors, for instance, the definition of the boundary layer momentum thickness. The unit of the boundary layer momentum thickness is also not provided. 

Author Response

Reviewer #2

Thank you very much for reviewing our manuscript. We really appreciate your good comments and suggestions that have significantly improved our manuscript. We have carefully considered all your comments and suggestions in the revised version. Please kindly find the response in the attached file.

Reviewer 3 Report

The authors failed to "fully" address my comments, which shows SLOPPINESS. Hence the manuscript CANNOT be published in the present format. Specifically:

1-  Table 1 (T Grid Parameters) did not properly show the sensitivity analysis. Please edit this table such that it shows the "optimum" grid size. Also, add the "core subroutine" of your ANSYS code (in 10-15 lines) in the appendix of your manuscript. 

2- "Galerkin" and "Bezier" numerical methods can be used to numerically solve the viscous resistance and inertial resistance coefficients mentioned in your paper. Please introduce and reference these numerical methods in your manuscript (look at my previous comments). 

3- Grammatical mistakes are abundantly found in your manuscript, find and correct all of them. For example:

* Line 35: "were investigated which combined with a control" (put a comma before "which")

* Line 40: "spanwise correlation lengthscale" ("lengthscale" to be changed to "length scale")

* Line 42: "above-mentioned contributions on airfoil noise" (contribution "to")

* Line 43: "also showed a great potential on noise" (a great potential "to")

Author Response

Dear  Reviewer,

Thank you very much for reviewing our manuscript. We really appreciate your good comments and suggestions that have significantly improved our manuscript. We have carefully considered all your comments and suggestions in the revised version. Please kindly find the response in the attached file.

Round 3

Reviewer 2 Report

Review report for “Numerical study on flow and noise characteristics of a NACA0018 airfoil with a porous trailing edge”, Sustainability-1922874

The improvements maded for this article is obvious.  There are some points needs to be revised:

1.  In the abstract "On the other hand, the loss of lift force is about 0.8%-1.5% at an angle of attack in the range of 0~10°. However, the drag force is increased to about 2%~17% for the suction-side-porous 25 trailing edge and 8%~24% for the fully-porous trailing edge.". However, the cotent corresponding to these conclusions has been deleted in this version. Therefore, revise the upper sentences (or just delete these conculions due to no support content in the article) in the abstract. 

2. For figure 11, add the unit for boundary layer  δ  nominal thickness and 
displacement thickness δ * .

3. For Section 3.2, "The suction side-filled airfoil has smaller area closed by the Cp curve before x/c=0.8 (c is the chord length), which will lead to a smaller lift."  It is suggested to provide a quantitative integral of the area closed by the Cp curve from x/c=0~1.0 to support this.

Author Response

Dear Reviewer:

Thank you once again, below are the improvements we have performed:

  1. In the abstract "On the other hand, the loss of lift force is about 0.8%-1.5% at an angle of attack in the range of 0~10°. However, the drag force is increased to about 2%~17% for the suction-side-porous 25 trailing edge and 8%~24% for the fully-porous trailing edge.". However, the cotent corresponding to these conclusions has been deleted in this version. Therefore, revise the upper sentences (or just delete these conculions due to no support content in the article) in the abstract. 

Answer: Thank you for suggestion, this part was deleted in the abstract.

  1. For figure 11, add the unit for boundary layer  δ nominal thickness and 
    displacement thickness δ * .

Answer: The unit is added in Figure 11.

  1. For Section 3.2, "The suction side-filled airfoil has smaller area closed by the Cp curve before x/c=0.8 (c is the chord length), which will lead to a smaller lift." It is suggested to provide a quantitative integral of the area closed by the Cp curve from x/c=0~1.0 to support this.

Answer: Thank you for the comment. We created an extra table to support the finding. From the integrated lift coefficients listed in Table 2, it is obvious that the lift force created by the solid airfoil is the largest, while the lift obtained from the suction-side-filled airfoil is larger than that from the full-filled airfoil.

Table 2. The lift coefficients compared at different angles of attack.

αo

  Solid

Fully-filled

Suction-side-filled

0

-4.55E-06

4.61E-05

0.17253

2

0.20126

0.19894

0.19989

4

0.38778

0.38142

0.38845

6

0.56762

0.55526

0.56973

8

0.74270

0.73436

0.73925

10

0.88547

0.86027

0.87523

Reviewer 3 Report

The authors addressed my comments.

Author Response

Dear Reviewer, thank you once again!

Best regards,

The authors